# The Role of Osteopontin in Respiratory Health and Disease

**DOI:** 10.3390/jpm13081259

**Published:** 2023-08-14

**Authors:** Georgios I. Barkas, Ourania S. Kotsiou

**Affiliations:** 1Department of Human Pathophysiology, Faculty of Nursing, University of Thessaly, 41500 Larissa, Greece; 2Department of Respiratory Medicine, Faculty of Medicine, University of Thessaly, 41110 Larissa, Greece

**Keywords:** respiratory disease, osteopontin, inflammation, cancer

## Abstract

The biological functions of osteopontin (OPN) are diverse and specific to physiological and pathophysiological conditions implicated in inflammation, biomineralization, cardiovascular diseases, cellular viability, cancer, diabetes, and renal stone disease. We aimed to present the role of OPN in respiratory health and disease. OPN influences the immune system and is a chemo-attractive protein correlated with respiratory disease severity. There is evidence that OPN can advance the disease stage associated with its fibrotic, inflammatory, and immune functions. OPN contributes to eosinophilic airway inflammation. OPN can destroy the lung parenchyma through its neutrophil influx and fibrotic mechanisms, linking OPN to at least one of the two major chronic obstructive pulmonary disease phenotypes. Respiratory diseases that involve irreversible lung scarring, such as idiopathic pulmonary disease, are linked to OPN, with protein levels being overexpressed in individuals with severe or advanced stages of the disorders and considerably lower levels in those with less severe symptoms. OPN plays a significant role in lung cancer progression and metastasis. It is also implicated in the pathogenesis of pulmonary hypertension, coronavirus disease 2019, and granuloma generation.

## 1. Introduction

Osteopontin (OPN) or secreted phosphoprotein 1 (SPP1) is a member of the SIBLING small integrin-binding ligand, N-linked glycoprotein (SIBLING) family of proteins, which map to human chromosome 4 [1(BMGC1) (BMGC2)]. OPN is a multifunctional glycoprotein and is found in numerous tissues of the body, such as saliva, bile, the brain, bone marrow, endothelial cells, smooth muscle cells, and pancreatic ducts. In salivary and sudoriferous glands, it is also secreted into body fluids by epithelial cells, for example, airway epithelial cells [1,2,3].

OPN presents different post-translational modifications [4,5]. It plays important roles in inflammation, biomineralization, cardiovascular diseases, cellular viability, cancer, diabetes, and renal stone disease through various pathophysiological mechanisms [6].

In biomineralization, OPN acts as an important communication mediator between osteoblasts and osteoclasts while playing a crucial role in the latter [6]. OPN acts as a conductor by playing a pivotal role in secretion levels of interleukin (IL)-10, (IL-1), (IL-3), interferon-γ (IFN-γ), integrin αvB3, and nuclear factor kappa B (NF-kB) by regulating osteoclast function and affecting CD44 receptors.

OPN is known to be a chemical attractant for macrophages and T cells, and research has been conducted to investigate its role in inflammation [2,6,7]. T cells secrete OPN when activated, affecting, particularly macrophages and causing them to migrate to sites of inflammation, therefore leading to higher levels of the protein in the serum of patients with systemic or chronic inflammation [7,8].

Moreover, OPN is a key cytokine in wound repair, serving as a chemotactic molecule to recruit inflammatory cells to the site of injury, promoting wound healing [8]. OPN communicates with cells via integrins and CD44, which involves an intracellular interaction where OPN, through a conserved region that separates both the integrin and CD44 binding domains, possesses distinct signaling functions that can link it to various forms of metastasis [9].

The biological functions of OPN are diverse and specific to a spectrum of physiological and pathophysiological conditions of the respiratory system [9,10]. OPN is constitutively expressed in the airways during healthy conditions. Evidence shows it is implicated in chronic inflammation associated with infection and regulates host immunity. High levels of OPN can be detected in sputum during states of disease characterized by prolonged airway inflammation, i.e., chronic obstructive pulmonary disease (COPD), cystic fibrosis (CF), and asthma [11]. OPN is implicated as being harmful in the case of systemic inflammation [12]. However, studies have demonstrated some beneficial roles for the protein, including protection against lung injury [11] and against cardiac ischemia-reperfusion injury [13]. Moreover, studies reported that it regulates cancer development. More specifically, in non-small cell lung cancers (NSCLCs), OPN induces vascular endothelial growth factor (VEGF) expression and facilitates disease progression [10].

This review aims to comprehensively present the role of OPN in respiratory health and disease.

## 2. Methods

We present data on the normal functions of OPN and its implications for various pulmonary diseases such as asthma, COPD, idiopathic pulmonary fibrosis, lung cancer, etc., while also discussing its mechanisms and physiological functions regarding disease severity. In that context, we conducted a literature search in the PubMed search engine (https://pubmed.ncbi.nlm.nih.gov (accessed on 10 June 2023)). The first search used the terms (Osteopontin OR OPN OR SPP1 AND respiratory diseases), and the following filters were applied (publication limit for the last five years and English language); the search resulted in 119 publications. Following that, we conducted a manual review of the top 50, sorted by date, and sorted the others by relevance. After thoroughly studying titles and abstracts, 82 were selected for full-text review. Of the 82 manually reviewed articles, all of them were screened from reference lists, which were also reviewed. Figure 1 presents the flowchart of the article selection process.

## 3. Mechanisms and Physiological Functions of Osteopontin

OPN is an acidic arginine–glycine–aspartate-containing adhesive glycoprotein that plays a role as an intrinsic component of the immune system [9]. OPN is rich in aspartic acid and consists of 300 amino acids [1]. OPN has two terminal zones, including the N-terminal and C-terminal, which bind two heparin molecules and CD44 variants, whereas the N-terminal includes integrin receptor binding zones [13]. The post-translational modifications (PTMs) of OPN, such as glycosylation and phosphorylation, significantly affect its structure and biological properties. For example, a reduction in sialylation may prevent OPN from binding to cell surface receptors [5].

OPN is important in tissue repair while also being associated with cellular regeneration. Moreover, OPN is involved in the proliferation of vascular smooth muscle cells and glomerular mesangial cells induced by hypoxia [3]. OPN expression is linked with fibrosis and scarring of the tissue, with upregulation of the protein found in animal models of renal disease.

OPN expression has been highly associated with numerous respiratory diseases, with high protein levels found in the airways and tissues of affected lungs and upregulation being present in more severe disease cases [14].

Many factors influence OPN release via various signal pathways, including protein kinase C (PKC) on specific cell types. PKC, for example, inhibits OPN release in Src^−/−^ fibroblasts stimulated by epidermal growth factor. Furthermore, increased OPN released in response to injury and illness is linked to cytokines [3].

OPN is a cytokine with diverse roles in tissue remodeling, fibrosis, immunomodulation, inflammation, and tumor metastasis [15]. Patients with OPN deficits have been reported to be protected from airway remodeling and hyperresponsiveness [15,16,17]. Furthermore, OPN can polymerize with itself or other extracellular matrix proteins, including fibronectin, via the catalytic action of tissue transglutaminase 2 (TGM2) [14].

Recent in vitro and animal investigations reveal OPN polymerization and support the idea that OPN causes neutrophils to gain chemotactic activity via its interaction with α9β1 integrin [14], providing insight into OPN’s polymeric function in human airways.

Moreover, a significant association between polymeric OPN and the concentration of alveolar macrophages in bronchoalveolar lavage (BAL) fluid was found, which suggests a chemotactic role of polymeric OPN for alveolar macrophages in the airways. This is consistent with studies supporting OPN’s role as a chemo-attractant of inflammatory cells such as macrophages [14,15].

OPN has been identified as an asthma biomarker, usually associated with the neutrophil asthma phenotype and indicating disease severity [18].

OPN plays a crucial role in the context of the airways. It exhibits high expression within the airways and affected lung tissues, particularly in more severe respiratory conditions. OPN is involved in processes such as tissue remodeling, fibrosis, immune modulation, and inflammation. It can form polymers and interact with other proteins in the extracellular matrix, resulting in chemotactic activity that attracts neutrophils and inflammatory cells to the airways. Moreover, OPN has been identified as a valuable biomarker for asthma, specifically associated with the phenotype characterized by neutrophilic inflammation, serving as an indicator of disease severity.

## 4. The Role of OPN in Bronchial Asthma

Asthma is a chronic inflammatory disorder of the conducting airways that can obstruct airflow [19,20,21]. Airway inflammation in asthma is a multicellular process associated with structural alterations of the airway components [19]. Inflammation of the lower airway most likely arises from a combination of factors such as genetic predisposition, environmental exposures, and previous respiratory disorders. In most cases, the inflammation present is type 2 inflammation, named for the type 2 T-helper cell lymphocyte. This type of inflammation is most common in allergy diseases and eosinophilic disorders. It is important to note that patients that do not present with a strong bias toward type 2 inflammation seem to exhibit a poor response to corticosteroids, and management tends to be more complex [19,20,21].

The chronic inflammation of the airway does contribute to airway remodeling, mainly through constant edema and mucus secretion. At the same time, certain disorders such as subepithelial fibrosis, angiogenesis, and mucous gland hyperplasia can result in permanent structural changes [22].

Airway remodeling can be defined as a set of changes in the composition, content, and organization of the cellular and molecular constituents of the airway wall [23,24]. The important structural alterations of the airway that happen in patients with chronic asthma are key to the severity and course of the disease. Airway remodeling affects both small and large airways. It is mainly characterized by subepithelial matrix deposition and fibrosis, with hyperplasia of airway smooth muscle cells. Overexpression of angiogenic factors occurs almost every time they are present. Increased deposition of extracellular matrix (ECM) is a characteristic of asthmatic airways and contributes to cell thickening and air obstruction [20,25]. When ECM is accumulated in the lung, it alters the function and structure of the tissue mainly because of the collagen fibers, fibronectin, and tenascin, which are abundant in ECM [25]. While airway smooth muscle cells also seem to produce increased amounts of collagen and fibronectin, resulting in a similar function to ECM.

OPN plays a vital role in chronic airway remodeling and bronchial hyperresponsiveness. OPN protein expression and its correlation have been extensively investigated over the years, with inconsistent results [21]. OPN is known to act as both an extracellular matrix molecule and a cytokine [21]. OPN is upregulated and remains a crucial component in allergen-induced airway remodeling in mouse models of asthma [21,26,27].

The effect of OPN on eosinophilic airway inflammation is known. In patients with severe refractory asthma, OPN levels were shown to be associated with increased levels of transforming growth factor-β1 (TGFβ-1). In contrast, eosinophil-derived OPN was shown to contribute to fibrosis through the IL-33/amphiregulin (Areg)/epidermal growth factor receptor (EGFR) axis in the context of Th2 inflammation [28]. Eosinophils have traditionally been considered the primary mediators of epithelial damage. They mediate epithelial damage via preformed effector molecules’ release [29]. Several studies have shown a significant correlation between OPN protein level and the number and activity of eosinophils in the sputum of patients with eosinophilic asthma, suggesting the potential role of OPN in attracting eosinophils into asthmatic airways [29,30,31]. The main studies reporting the role of OPN in bronchial asthma are presented in Table 1.

## 5. Role of OPN in Chronic Obstructive Pulmonary Disorder

Chronic obstructive pulmonary disease (COPD) is characterized by persistent respiratory symptoms and progressive airflow obstruction that is not reversible [32]. COPD is associated with both airway and systemic inflammation. Inflammatory markers such as OPN seem higher in more severe disease cases [33]. The main risk factor for COPD is tobacco smoking, with most patients suffering from the disease being active, passive, or former smokers [34,35]. However, long-term exposure to lung irritants such as air pollution, dust, and other rare genetic conditions may also be causal factors of the disorder [32]. The pathological features of COPD are pulmonary inflammation, remodeling changes in the airways, fibrosis, and tissue injury [36,37,38,39,40].

OPN plays an important role in inflammation via the recruitment of neutrophils and tissue remodeling [41]. COPD is characterized by remodeling of the airway epithelial lining and increased mucus discharge. It has been shown that cigarette smoke increases OPN production in the airways; consequently, OPN levels are high in COPD patients [41]. Studies confirmed the upregulation of OPN expression in COPD patients and its association with shorter survival time, while it is still unclear whether OPN can be a biomarker for the disease [42].

Epithelial cells of the small airways, which play an important role in the progression of COPD, express very high amounts of OPN in patients with severe forms of the disease and acute exacerbations. While the same remark has been made on the sputum of the patients [41,42]. OPN contributes to inflammation by promoting an influx of neutrophils, prolonging respiratory inflammation, and helping in airway remodeling and chronic inflammation [33,36,39,40,41,42]. Chronic lung inflammation is key to the pathogenesis of COPD and causes most clinical manifestations. OPN’s function as a cytokine certainly exacerbates inflammatory and profibrotic functions [40,43]. Patients were studied on a small scale to determine the OPN plasma levels and their differences in COPD and AECOPD (Acute exacerbations of COPD), which resulted in patients with AECOPD having much higher levels of OPN in their plasma [43]. However, there was not a significant decrease in recovery, but their results cannot be fully trusted mainly because of their small number of test subjects. Another study found increased levels of OPN in sputum supernatant in patients with COPD compared with healthy smokers and nonsmokers, with a follow-up regression analysis resulting in a significant association between OPN and sputum neutrophils, therefore highlighting the chance of OPN playing a role in the pathogenesis of emphysema [36].

Airway remodeling contributes to a smaller airway lumen and increased alveoli size, while increased sputum secretion attenuates the problem. OPN is upregulated in acute exacerbations of COPD, while its functions on higher levels of ECM contribute to fibrosis in various lung tissues, reducing their elasticity.

OPN is found at higher levels in patients suffering from COPD and AECOPD, giving us insights into its function on the disease. In comparison, patients with AECOPD have much higher OPN plasma levels, which can be attributed to the site’s inflammation or the smoking habit since smoking leads to higher OPN levels.

In summary, OPN plays an important role in COPD either through its inflammatory functions or fibrotic mechanisms; OPN is overexpressed in COPD lungs, leading to constricted and obliterated airways. Whereas OPN levels are linked to a poorer prognosis and less survival of patients, therefore limiting its potential use as a biomarker. OPN can destroy the lung parenchyma through its neutrophil influx and fibrotic mechanisms, linking OPN to treatable traits of COPD. The main studies reporting the role of OPN in COPD are presenting in Table 2.

## 6. OPN in Idiopathic Pulmonary Fibrosis

Idiopathic pulmonary fibrosis (IPF) is a destructive lung disease characterized by rapid progression leading to lung fibrosis [44]. It is the most common idiopathic interstitial pneumonia (IIP), accounting for 25–30% of patients diagnosed with IIP in North America and Europe [45,46,47]. Even though the exact mechanisms of fibrosis remain unknown, in most patients, previous injuries to the respiratory system combined with genetically predisposed alveolar epithelium that are followed by a failure to re-epithelialize and repair are usually the main reasons [44,45,46,47,48,49]. Fibrosis is defined as excessive, pathologic deposition of ECM during wound healing. Fibrogenesis is a highly orchestrated process that integrates multiple cell types and signaling mechanisms across organ systems. It elicits an inflammatory response to specific triggers, such as infections and wounds, that recruits fibroblasts and activates a subset of cells, myofibroblasts, to deposit ECM in the form of collagen and other proteins. In a normal state, this mechanism would resolve with apoptosis of the myofibroblasts, while in fibrotic disease, the profibrotic activators and myofibroblasts persist, causing [48]. IPF leads to scarring of the lung parenchyma, which results in reduced quality of life and earlier mortality. It is an age-related disorder, with patients having a life expectancy of 3–5 years after diagnosis if left untreated [49].

OPN is involved in various biological processes, including inflammation and tissue fibrosis; it is known to be involved in various biological functions and can exert cytokine-like functions in promoting cell adhesion, migration, and growth of various cell types. OPN is upregulated during inflammation processes and fibrotic organ and tissue remodeling and has been associated with different diseases [50,51,52]. OPN plasma levels have been associated with several inflammatory diseases, including cancer, wound healing, and cardiovascular diseases such as pulmonary arterial hypertension [51]. In contrast, high levels accelerate fibrosis progression by mediating fibroblast proliferation and regulating the expression of matrix metalloproteinase (MMP) to promote the accumulation of ECM [50]. SPP1 (OPN) is a consistently observed marker of IPF; it is a highly selective marker for an expanded subpopulation of macrophages found in human IPF.

The SPP1 gene is strikingly deposited In fibrotic IPF lower lobes, associated with fibroblastic foci [53]. It is important to note that OPN was localized to alveolar epithelial cells in IPF lungs and significantly elevated in bronchoalveolar lavage for IPF patients. OPN has been proven to be mainly expressed by alveolar epithelial cells; therefore, the integrin receptors for OPN are widely expressed in lung epithelial cells and fibroblasts [54].

In summary, OPN plays a vital role in IPF, a progressive and lethal disorder characterized by proliferation and excessive accumulation of extracellular matrix in the lung. OPN is involved in tissue fibrosis and can accelerate the progression of fibrosis through different and not yet fully understood mechanisms. Higher OPN levels were present in IPF patients, while the OPN gene SPP1 is a known and observed biomarker for IPF and is deposited in fibrotic lungs. Therefore, OPN is identified as one of the genes distinguishing IPF from normal lungs. The main studies reporting the role of OPN in IPF are reporting in Table 3.

## 7. OPN in Cancer Genesis and Non-Small Cell Lung Cancer

Lung cancer is a molecularly heterogeneous disease, and to understand the role of OPN in it, we first need to understand its mechanisms. Smoking is the most common risk factor for developing lung cancer, and it has been shown that lung tumors harbor smoking-related genetic signatures. Lung cancer comprises cells with distinct molecular functions and features, resulting in intratumoral heterogeneity [55,56].

Tumors within the non-small cell lung cancer (NSCLC) category have been identified as genetically varietal, leading to many oncogenic mechanisms. The two most common subtypes of NSCLC, adenocarcinoma (ADC) and squamous cell carcinoma (SCC), display an overlap between the causing genes and cellular pathways, though significant differences exist [56].

Tumor angiogenesis is the formation of new vasculature to supply nutrition and oxygen from pre-existing blood vessels to tumors and is vital to cancer survival and invasion. Most tumors overexpress VEGF-A and its receptors [57]. VEGF is an angiogenic inducer and a mediator of pathological angiogenesis while being linked to various malignancies. It also promotes the growth of vascular endothelial cells and is a survival factor [58]. Cancer progression depends on the accumulation of metastasis-supporting genes and physiologic alternating signals regulated by cell signaling molecules such as ECM. Those signaling molecules contribute to the interaction between cancer and endothelial cells, which plays an important role in the development of cancer invasion [59]. The ECM molecule is the key to tumor progression and metastasis, making it a causal agent for poorer progression and a stronger invasion of lung cancer.

OPN is an ECM ligand for integrins and is highly expressed in osteoblasts and osteoclasts. It is a key factor in biomineralization and contributes to various metastasis-associated mechanisms, including proliferation, survival, adhesion, migration, invasion, and angiogenesis [54]. Up-regulation of the protein is usually associated with poor prognosis, severities, and, overall, more difficult therapeutic actions [4,15,16,18,19]. OPN has been demonstrated to play a role in the metastasis of non-small cell lung cancer (NSCLC) [54]. NSCLC accounts for over 80% of all diagnosed lung cancer cases. While lung cancer is the leading cause of cancer death worldwide [60]. Most NSCLC cells overexpress VEGF-A, a pivotal factor in tumor angiogenesis.

Even though specific therapies that target the neutralization of VEGF-A have improved the disease’s survival rate, the median survival of patients is shorter than 18 months [60,61]. OPN induces VEGF expression and facilitates disease progression and inflammatory regulation, contributing to the disease [7]. OPN has been identified as a robust biomarker of tumor progression and metastasis. It has been reported that the levels of OPN correlate with tumor grade and prognosis in patients with various cancers [62,63,64]. Unfortunately, the evidence of oncogenicity in NSCLC is sparse, with newer research now linking higher protein expression with lower patient survival in patients suffering from NSCLC [64]. Several studies revealed that the value of OPN levels matters in the prognosis of lung cancer and can increase the metastatic potential while promoting pathological responses [62,63,64,65]. It has been proven that the SPP1 gene (OPN) can inhibit the autophagy and apoptosis of cells, leading to the development of lung cancer cells and promoting cancer cell proliferation [66]. It is present and highly expressed in lung cancer cell lines and tissues compared with normal human lung epithelial cells but does not affect bronchial epithelial cell proliferation [61,62,63,64,65,66]. In conclusion, we can identify the significant role OPN plays in cancer and tumor genesis and its presence and importance in NSCLC. The VEFG-A factor, which is critical in angiogenesis, can be expressed from OPN, and, when present, OPN can facilitate tumor genesis, worsen prognosis, and lower patient survival overall.

In summary, OPN plays a significant role in lung cancer progression and metastasis. It is highly expressed in lung cancer cells and tissues, particularly NSCLC. OPN promotes various metastasis-associated mechanisms, including proliferation, survival, adhesion, migration, invasion, and angiogenesis. Its upregulation is associated with poor prognosis and more aggressive disease. OPN induces the expression of VEGF, a key factor in tumor angiogenesis, contributing to disease progression. OPN levels correlate with tumor grade and prognosis in lung cancer patients, and it inhibits autophagy and apoptosis, promoting cancer cell proliferation. Overall, OPN is a biomarker of tumor progression and metastasis in lung cancer. The main studies reporting the role of OPN in lung cancer are reporting in Table 4.

## 8. OPN in Pulmonary Hypertension

Even though pulmonary hypertension (PH) is not a strictly respiratory disorder but rather a cardiac one, it affects the respiratory system significantly. It has an impact on the breathing quality of the patient [67].

PAH is defined as the mean pulmonary arterial pressure of 20 mmHg or greater at rest [67,68]. It is a chronic hemodynamic and pathologic disease characterized by progressive remodeling of the distal pulmonary arterioles that results in right ventricular failure, pulmonary vascular resistance (PVR), and eventually death [69,70,71]. Disturbed endothelial cell proliferation, smooth muscle cell hyperplasia, and chronic inflammation of the pulmonary vessels are important mechanisms contributing to the pathogenesis of idiopathic PAH and secondary PH [72]. Those specific mechanisms are either curated or contributed by OPN, as OPN plays a role in several forms. Pulmonary vascular remodeling in PAH is characterized by the thickening of adventitia, media, and intima caused by hypertrophy and hyperplasia of the cell type within each layer of tissue. The cell types that undergo those changes are mainly fibroblasts, endothelial cells, and pulmonary arterial smooth muscle cells [70]. Other than hypertrophy and hyperplasia, the cells also exhibit an increased deposition of ECM—an important function of OPN—as well as fibronectin and collagen [70].

OPN is an essential contributor to fibrotic elements, with activation of the phenotype of PASMCs (Highly proliferating, migratory) and pulmonary adventitia fibroblasts in hypoxic PAH [69]. It is expressed and upregulated during inflammation and expressed in high levels in patients with fibrotic lung disorders, and it has been demonstrated that OPN is one of the most upregulated genes in the lungs of patients with severe PAH, while the SPP1 gene shows a strong correlation with mPAP (mean pulmonary arterial pressure), which should be less than 20 mmHg [70]. Certain studies have proven that patients with CTEPH (chronic thromboembolic pulmonary hypertension) had higher concentrations of OPN compared to patients with pulmonary embolism and that OPNs downstream target (MMP-9) are present in profibrotic areas of CTEPH tissue material [51].

OPN levels in patients with IPAH (idiopathic pulmonary arterial hypertension) are elevated compared to control subjects [72]. And OPN has been classified as a potential biomarker for the disease [70,71,72,73]. OPN can be an independent prognostic factor in terms of the outcome. It plays a role in myocardial stress response and fibrotic genesis leading to epithelial damage and implications on the arterial supply [51,72].

OPN was found to be upregulated in several heart failure models connected to the arterial and pulmonary systems. OPN plasma levels are elevated in patients with left-sided heart failure—a disorder heavily linked to PAH—and correlate with an adverse prognosis [74].

The underlying molecular mechanisms that initiate change in PH remain unknown, but OPN is essential in initiating obstructive remodeling of the pulmonary vessel wall [70,74]. OPN correlates with impaired proper ventricular function and remodeling in patients, implying an important role in myocardial remodeling in response to biomechanical stress [51]. OPN is a key mediator expressed by pulmonary vascular cells and contributes to pulmonary hypertension while being overexpressed in COPD and PH patients [75].

PH is a chronic disease characterized by a poor prognosis; OPN plays an important role in the disease’s genesis, progression, and prognosis. In PH, epithelial cell damage, fibrosis, remodeling, and hyperplasia are critical to its prognosis. At the same time, all of those functions are upregulated by OPN, linking the protein to the disease. Several studies have found overexpressed levels of OPN in plasma, tissue of the impaired lungs, and arterial vessels. It proves its correlation with the disease and gives hope that OPN is a promising biomarker. The main studies reporting the role of OPN in pulmonary arterial hypertsnsion are presented in Table 5.

## 9. OPN in Multiple Sclerosis

Multiple sclerosis (MS) is an organ-specific T-cell-mediated autoimmune disease that affects young adults in large numbers, making it the most common non-traumatic disabling disease [76]. MS affects approximately 2.3 million people worldwide, with it being most prevalent in North America (140 cases per 100,000) and Europe (109 cases per 100,000) [77].

MS is an inflammatory disease of the central nervous system (CNS) with an unknown etiology that causes many symptoms across several organ systems involving the motor, sensory, visual, and autonomic systems.

Even though MS is not primarily a respiratory disease, it predominantly affects the respiratory system through its muscular strength and endurance limitations, affecting the functionality of respiratory muscles vital to normal breathing and lung capacity.

While other respiratory functions, such as coughing—a mechanism of protection from outside particles and bacteria—are primarily affected, leading to lower respiratory immunity and respiratory diseases such as pneumonia [78].

MS is an autoimmune disease highly linked with the inflammatory response and T cells; therefore, a correlation between the disease and OPN is factual.

OPN acts as a chemokine and has integrin properties; it is also secreted by macrophages [1]. Therefore, by binding to those receptors, OPN regulates immunologic responses, developmental processes, and tissue remodeling. It has been shown that immune cells can produce OPN, while excessive OPN expression has been linked to several disorders, such as inflammation, fibrosis [79], and MS [80].

The connection between OPN and MS has been studied, as there is a relationship between OPN expression and symptom severity in MS patients [81]. The upregulation of OPN promotes T-cell death through the expression of pro-apoptotic proteins, neurological relapse in MS patients, and plaque deposited in the brains of patients [79].

In systemic sclerosis, higher amounts of OPN in serum have been found, suggesting an increase due to the inflammatory process in the disease [82].

OPN is found in the CNS and is upregulated in lesions in patients suffering from MS. Furthermore, studies on mouse models with OPN deficiency conclude the protective nature of the disorder because of the less severe phenotype of MS they exhibit [83].

As mentioned, higher levels of OPN in MS patients have been found, either in the blood or CSF (cerebrospinal fluid). The levels corresponded with the severity of the disease, demonstrating the link between OPN and MS and its potential use as a biomarker for the disease [84].

MS is a severe autoimmune, degenerative disease that primarily affects young people and causes disability and chronic pain. OPN has a clear link to the disease, with higher levels in both spine fluid and peripheral blood in patients with severe forms of the disease, leading to its use as a potential prognostic biomarker.

OPN is found in the CNS and, when overexpressed, causes neurological symptoms and triggers severe forms of MS, affecting CNS tissues and causing T-cell death and cell apoptosis.

This contributes to faster neurodegeneration and the emergence of neurodegenerative diseases such as Alzheimer’s. It is important to note that even though the impact of OPN is mainly catastrophic, there is hope of it being used as a target for therapy or as a way to slow disease progression.

MS is a severe degenerative disease affecting many people, causing chronic pain and severe systemic inflammation. OPN, a protein involved in immune responses and tissue remodeling, has been implicated in multiple sclerosis (MS). In MS patients, higher levels of OPN have been found to correlate with disease severity. OPN is upregulated in CNS lesions in MS patients and has been linked to T-cell death and neurodegeneration. Its potential as a biomarker and therapeutic target for MS is being explored. The main studies reporting the role of OPN in multiple sclerosis are presented in Table 6.

## 10. OPN in COVID-19

Severe acute respiratory syndrome coronavirus 2 (SARS-CoV-2) is a highly transmissible and pathogenic coronavirus that first emerged in late 2019 and caused a worldwide pandemic called coronavirus disease 2019 (COVID-19). The studies of OPN and its correlation with the disease or the SARS-CoV-2 virus are relatively limited.

Albeit OPN, a cytokine-like matrix-associated protein, drives the expression of furin, a proprotein that has an essential role in enhancing the infectivity of the virus by promoting its entry and replication in the host. It has been noted that diabetes patients with increased furin had worse outcomes than ones without diabetes, therefore pointing to a connection between the furin–OPN axis and its correlation to worse outcomes [85].

In rare cases, the immune system reaction to SARS-CoV-2 infection can lead to a cytokine storm syndrome that substantially increases mortality risk [86,87]. OPN levels of patients admitted to the ICU with acute and severe forms of COVID-19 were upregulated, given that pro-inflammatory and cytokine properties of OPN affect the respiratory system when the virus is modulating inflammation in the host [87]. It has been found that circulating levels of OPN were elevated in patients with COVID-19 upon hospital admission [88]. The concentrations of OPN directly reflect disease severity and represent an independent risk factor for the clinical course [87,88]. OPN is a T-helper cytokine and is thus believed to be a regulator of inflammation in the respiratory system in several diseases [89].

Since OPN is a T-helper cytokine that promotes inflammation, its link to COVID-19 is highly possible, while COVID-19 patients with severe symptoms admitted to the ICU and assessed with laboratory work were found to have increased ferritin and OPN and enhanced inflammation and fibrogenesis [90]. The enhancements are confirmed by OPN and ferritin in both inflammatory and fibrotic lesions. This fact associated OPN with the disease severity of COVID-19 even more. When assessed for its biomarker properties for COVID-19 and multisystem inflammatory syndrome (MIS-C) in children, OPN levels were significantly elevated in children hospitalized with moderate or severe forms of COVID-19 and MIS-C compared to the levels in mild or asymptomatic patients. OPN also differentiated from other inflammatory markers in different forms depending on the severity of the disease, while other markers did not [91].

OPN is a T-helper cytokine believed to be a regulator of inflammation in the respiratory system in several diseases. From a biomarker perspective, OPN can predict disease severity and is found in high levels in coronavirus patients with severe forms of the disease. However, it is not a viable biomarker of the disease, just its prognosis. Plasma levels of patients admitted to the ICU with COVID-19 and acute respiratory disorders show severe inflammation and fibrotic tissue in the airways. At the same time, OPN in the same sites is overexpressed. OPN is involved in biomineralization, cellular viability, and diabetes. While diabetic patients who have higher amounts of ferrin also seem to exhibit higher amounts of OPN in COVID-19. OPN acts as an organism by playing a pivotal role in the secretion levels of interleukin-10 (IL-10), interleukin-12 (IL-12), interleukin-3 (IL–3), interferon-γ (IFN-γ), integrin αvB3, nuclear factor kappa B (NF-kB), macrophages, and T cells, all organisms involved in respiratory diseases that are outcomes of severe coronavirus disease.

OPN acts as a biomarker for the severity of the disease, as there is overexpression in patients with severe forms of COVID-19. Since OPN is involved in inflammation, fibrosis, immune response, and ECM upregulation, it participates in COVID-19 pathogenesis. The main studies reporting the role of OPN in COVID-19 are presented in Table 7.

## 11. OPN in Tuberculosis

Tuberculosis (TB), caused by bacteria of the Mycobacterium tuberculosis (MTB) complex, is the leading cause of death worldwide by an infectious disease among adults. Even though tuberculosis epidemiological numbers are not as high as they were in previous centuries, the disease still has a high prevalence rate among low socioeconomic sections of the population, and according to the WHO, since the HIV/AIDS disease emerged, tuberculosis has been a major killer of the human population [92,93,94,95].

There is a correlation between plasma OPN levels and clinical parameters in patients with pulmonary TB, where OPN levels were significantly higher in sputum acid-fast smear-positive patients than in smear-negative ones. At the same time, the protein concentrations increased as the pulmonary lesion did, with patients with severe and time-advanced TB having the highest concentrations. This points to a correlation between plasma OPN concentration and the severity of TB [96].

OPN, a chemo-attractive cytokine, has been associated with granulomas—the main clinical phenotype of TB. TB lesions consisting of necrosis and lymphoid aggregation above alveolar air spaces were found to have OPN levels higher than usual [97].

OPN regulates macrophages and T-cell migration while helping activate and express cytokines in TB. Mediating the accumulation of macrophages, macrophage-derived epithelioid cells, and giant cells during granuloma formation [98].

T-cell immunity and its link with OPN have been discussed earlier in the paper. T cell immunity is critical in controlling TB infection, with T-cytokine production, interferon-γ (IFN-γ), being genetically responsible for the development of TB and OPN correlating with it [97,98].

IFN-γ is an important mediator of macrophage activation critical in MTB. In OPN-deficient mice, BLM (bleomycin)-induced lung inflammation and subsequent fibrosis were ameliorated [99], mainly because of the link between OPN and the Th1 response. While in TB lungs, sputum, and blood, there was an elevation of IFN-γ. A link between OPN and IFN-γ levels is clarified, with elevations or decreases in both being associated [100].

Pleural fluid OPN levels were higher in exudative pleural effusions (PEs) compared to transudative PEs. In contrast, the opposite result was found in plasma, with higher levels in transudatory patients and lower levels in exudative patients. OPN is significantly correlated with pleural inflammation, and endothelial cell mechanisms are therefore involved in the pathogenesis of exudative PE [101].

To summarize, OPN is a protein highly expressed in the tissues, blood, and sputum of people with TB, and its levels correlate inversely with the severity of the disease. OPN interacts with T cells to mediate the immune response, and cytokines control macrophage accumulation, leading to various granulomas. OPN can therefore be used as a biomarker for disease severity and could be a potential future therapeutic target to treat TB as it reduces disease severity and immune response. The main studies reporting the role of OPN in tuberculosis are presented in Table 8.

## 12. Conclusions

The role of OPN in a wide range of respiratory disorders was outlined in this review. The biological functions of OPN are diverse and specific to physiological and pathophysiological conditions implicated in respiratory health and disease. OPN influences the immune system and is a chemo-attractive protein correlated with respiratory disease severity. There is evidence that OPN can advance the disease stage associated with its fibrotic, inflammatory, and immune functions. OPN contributes to eosinophilic airway inflammation. OPN can destroy the lung parenchyma through its neutrophil influx and fibrotic mechanisms, linking OPN to at least one of the two major chronic obstructive pulmonary disease phenotypes. Respiratory diseases that involve irreversible lung scarring, such as idiopathic pulmonary disease, are linked to OPN, with protein levels being overexpressed in individuals with severe or advanced stages of the disorders and considerably lower levels in those with less severe symptoms. OPN plays a significant role in lung cancer progression and metastasis. It is also implicated in the pathogenesis of pulmonary hypertension, coronavirus disease 2019, and granuloma generation.

OPN is a protein resembling an extracellular matrix and is a crucial cytokine in Th2 and Th1 immune responses. OPN influences the immune system and works as a chemo-attractive protein, worsening disease severity. Most of the studies cited in this review determine the OPN levels in patients’ blood, tissue, or sputum, firmly tying the protein to potential as a biomarker. There is evidence that OPN can advance disease severity in a wide range of respiratory and autoimmune diseases, with its prominent role in the pathogenesis and clinical presentation of respiratory diseases being primarily associated with its fibrotic, inflammatory, and immune functions.

Although the role of OPN as a biomarker has been investigated in respiratory physiology and various diseases and linked to disease severity, its use in the clinical setting is limited mainly because it lacks specificity—it is a diverse protein with a wide range of actions.

We succeeded in elucidating the function of OPN in numerous disorders. We concluded that OPN plays a significant role in several conditions where it functions as a chemokine, has integrin characteristics, and is released by macrophages. Therefore, by binding to a wide variety of receptors, OPN controls immune responses, developmental processes, and tissue remodeling, making it a multifunctional protein. Future research directions regarding the role of OPN as a powerful, cost-effective, and rapid diagnostic, therapeutic, or prognostic biomarker in daily clinical practice are needed.

## Figures and Tables

**Figure 1 jpm-13-01259-f001:**
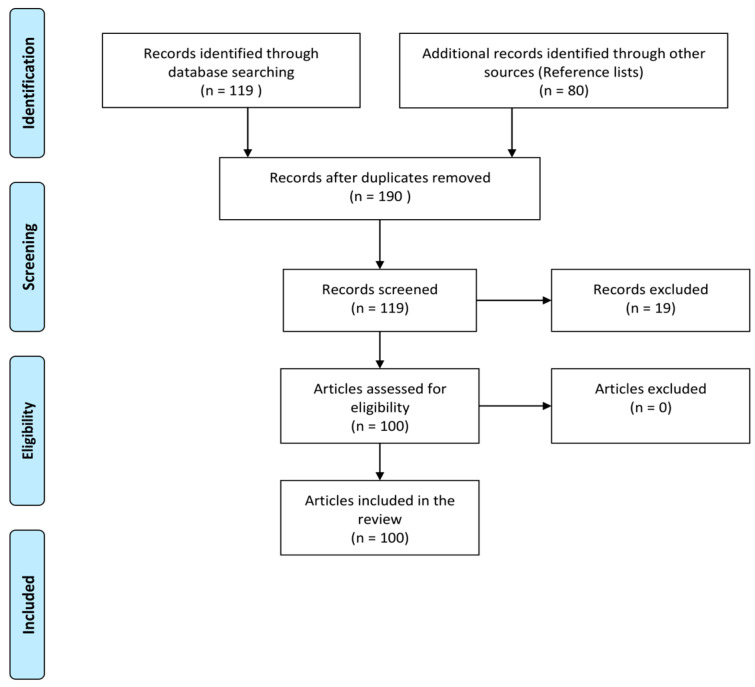
The flowchart of the article selection process.

**Table 1 jpm-13-01259-t001:** The main studies reporting the role of OPN in bronchial asthma.

Author/Ref	Study Design	Study Population	Main Findings
Samitas K et al. [19]	Observational study	54 asthmatic patients (35 with MMA and 19 with SA) and 17 non-atopic healthy controls were followed for ≥1 yr.	OPN was upregulated in bronchial tissue in asthmatics. Subepithelial inflammatory cells in SA patients expressed significantly more OPN than MMA patients, suggesting that OPN expression correlates with disease severity.
Kohan M et al. [26]	Experimental study	BALB/c mice of unknown number were sensitized and exposed to OVA or saline inhalations for 5 weeks and killed 24 h after the last inhalation.	OPN expression was up-regulated in lung tissue and BAL fluid of OVA-treated mice and correlated with collagen content and peribronchial smooth muscle area.
Kohan M-Breuer R et al. [27]	Experimental study	OPN^−/−^ (B6. CgSpp1tm1Blh/J) and C57Bl/6 mice. Ten mice per group were used.	Reduced airway remodeling in OPN^−/−^ mice.
Trinh HKT et al. [28]	Prospective observational	131 adults with asthma, including 48 with LOA and 83 with EOA, and 226 non-atopic healthy controls (HCs).	This study demonstrated the role of OPN as a central mediator in the pathogenesis of LOA rather than EOA.

Abbreviations: BALB/c, albino lab-bred mice; EOA, early-onset asthma; LOA, late-onset asthma; MMA, mild to moderate asthma; OVA, ovalbumin; SA, severe asthma.

**Table 2 jpm-13-01259-t002:** The main studies reporting the role of OPN in COPD.

Author/Ref	Study Design	Study Population	Main Findings
Papaporfyriou et al. [36]	Observational cross-sectional	47 patients with COPD and 40 healthy subjects (20 smokers) were studied.	OPN levels (pg/mL) were significantly higher in patients with COPD than healthy smokers and nonsmokers [median [interquartile range]; 1340 (601, 6227) vs. 101 (77, 110) vs. 68 (50, 89), respectively; *p* < 0.001].
Mou S et al. [40]	Prospective observational study	52 subjects with AECOPD + CAP and 93 subjects with AECOPD from two clinical lefts were enrolled.	Patients with AECOPD + CAP had increased sputum volume, sputum purulence, diabetes mellitus, and longer hospital stays than AECOPD patients (*p* < 0.05). While OPN, among other inflammatory biomarkers, was found upregulated in those patients as well.
Lee SJ et al. [43]	Observational study	Plasma OPN levels were measured and compared in patients with COPD exacerbation (*n* = 64), patients with stable COPD (*n* = 68), and healthy controls (*n* = 30).	Patients with COPD exacerbation had increased plasma OPN levels compared to stable COPD and healthy controls (32.6 ± 29.6, 17.6 ± 11.1, 8.4 ± 6.1 ng/mL, respectively; *p* < 0.001).

Abbreviations: AECOPD, acute exacerbations of chronic obstructive pulmonary disease; CAP, community-acquired pneumonia; COPD, chronic obstructive pulmonary disease.

**Table 3 jpm-13-01259-t003:** The main studies reposting the role of OPN in idiopathic pulmonary fibrosis.

Author/Ref	Study Design	Study Population	Main Findings
Moss et al. [48]	Review	No study population.	The review mentions results from scRNA-seq that demonstrate an increase in SPP1-expressing macrophages that localize to the fibroblastic foci.
Gui X et al. [50]	Prospective observational study	71 subjects (32 AE-IPF patients and 39 S-IPF patients), and 20 healthy controls.	Elevated serum OPN levels during acute exacerbation in IPF patients and OPN levels were positively correlated with CRP and LDH levels. Serum OPN predicted survival in IPF patients.
Hatipoglu et al. [52]	Prospective interventional study	8–10-week-old C57/BL6 male mice were used in this study.	OPN silencing protected from fibrosis in BLM-induced IPF model in mice. Therefore, OPN plays an important role in IPF, and its silencing is a promising therapy target.
Morse et al. [53]	Observational single-cell RNA sequencing	Healthy control lungs and IPF lung tissue were obtained under a specific protocol.	Genes upregulated in SPP1 macrophages in IPF lungs.

Abbreviations: AE-IPF, acute exacerbations of idiopathic pulmonary fibrosis; BLM, bleomycin; C57/BL6, C57 black 6 mice; CRP, C-reactive protein; LDH, lactate dehydrogenase; S-IPF, stable idiopathic pulmonary fibrosis; SPP1, secreted phosphoprotein 1 (OPN gene).

**Table 4 jpm-13-01259-t004:** The main studies reporting the role of OPN in lung cancer.

Author/Ref	Study Design	Study Population	Main Findings
Hu Z et al. [59]	Observational study	279 male and 41 female lung cancer cases.	OPN plasma levels were significantly higher in the NSCLC patients than those observed in patients with SCLC, patients with pulmonary benign diseases, and healthy donors.
Feng Y-H et al. [61]	Observational study	79 lung cancer patients.	Expression of Oct4 was positively correlated with Egr1 and OPN expression in human lung cancer.
Hao C et al. [64]	Observational study with both cross-sectional and longitudinal elements.	A total of 77 (paired tumor and adjacent normal tissues from donors.	The expression levels of RON had prognostic value, such as OPN levels, in patients with NSCLC.
Xu C et al. [65]	Cross-sectional observational study with a comparison group.	A total of 96 SCLC patients before and after first-line chemotherapy, compared to 60 healthy controls.	The serum OPN levels in SCLC patients before treatment were significantly higher than those of the healthy controls (*p* < 0.001). Serum OPN levels were correlated to disease stage, tumor size, and lymph node metastasis. The serum OPN levels were an independent predictor of overall survival.

Abbreviations: NSCLC, non-small cell lung cancer; Oct4, octamer transcription factor 4; Egr1, early growth response 1; RON, receptor tyrosine kinase; SCLC, small cell lung cancer.

**Table 5 jpm-13-01259-t005:** The main studies reporting the role of OPN in pulmonary arterial hypertension.

Author/Ref	Study Design	Study Population	Main Findings
Meng et al. [69]	Observational and interventional	Blood samples were obtained from apparently healthy subjects without signs of significant heart diseases (*n* = 24), patients with CHD but no PAH (*n* = 22), patients with CHD and severe PAH (*n* = 25), and patients with Eisenmenger syndrome (*n* = 20).	OPN levels were increased in lungs and plasma of patients with CHD/PAH.
Mura et al. [70]	Observational	14 patients with group I PAH, 1 patient with chronic thromboembolic PH, and 11 normal lung samples, obtained from the region of normal tissue flanking lung cancer resections in PH-free patients.	OPN gene expression was one of the most upregulated genes in the lungs of patients with severe PAH, and its gene expression strongly correlated with mPAP.
Lorenzen JM et al. [72]	Retrospective/prospective cohortstudy	A treatment-naive cohort (*n* = 70) and a prospective cohort (*n* = 25).	OPN levels in patients with IPAH were elevated compared to control subjects.
Rosenberg M et al. [74]	Prospective study	71 consecutive patients with PH and 29 patients with PH of various origin was independently collected.	Expression of Oct4 was positively correlated with Egr1 and OPN expression in human lung cancer.

Abbreviations: CHD, congenital heart defects; IPAH, idiopathic pulmonary arterial hypertension; mPAP, mean pulmonary arterial pressure; PH, pulmonary hypertension.

**Table 6 jpm-13-01259-t006:** The main studies reporting the role of OPN in multiple sclerosis.

Author/Ref	Study Design	Study Population	Main Findings
Xu et al. [79]	Review	No study population.	OPN was overexpressed in autoimmunity, and both secretory and intracellular types of this unique molecule had pathogenic roles in autoimmune diseases.
Gundogdu B et al. [82]	Observational cross-sectional	86 patients with SSc, 46 patients with SLE, and 38 healthy controls were enrolled in the study.	Serum OPN levels were higher in the SSc and SLE groups compared to the control group (*p* < 0.01 and *p* < 0.001, respectively).
Rosmus et al. [83]	Review	No study population.	OPN was significantly upregulated in MS lesions in comparison to healthy brain tissue and was found to be elevated in serum samples of MS patients.
Agah et al. [84]	Systematic review and meta-analysis	No study population.	Both peripheral blood and CSF levels of OPN were increased among MS patients compared to controls.

Abbreviations: CSF, cerebrospinal fluid; MS, multiple sclerosis; SLE, systemic lupus erythematosus; SSc, systemic sclerosis.

**Table 7 jpm-13-01259-t007:** The main studies reporting the role of OPN in COVID-19.

Author/Ref	Study Design	Study Population	Main Findings
Hayek SS et al. [89]	Observational study	341 hospitalized COVID-19 patients with plasma samples collected within 48 h of admission.	Circulating levels of OPN were elevated in patients hospitalized for COVID-19, while elevated levels of OPN were associated with an increased risk of death and need for mechanical ventilation.
Ueland T et al. [90]	Observational study	65 hospitalized COVID-19 patients were included with 195 serum samples.	Evaluation of biomarkers revealed that the PF4+ patients had higher ferritin and OPN levels during the first 10 days of admission, while ferritin and OPN remained higher after 3 months, with persistent elevation of OPN in PF4+ patients.
Reisner A et al. [91]	Retrospective study	26 children hospitalized for COVID-19 categorized into groups regarding the severity of symptoms.	The severe COVID-19 and MIS-C groups had significantly higher levels of OPN at time of collection (median = 430.31 and 598.11 ng/mL, respectively) compared to the other two groups (asymptomatic or minimally symptomatic, mild/moderate).

Abbreviations: MIS-C, multisystem inflammatory condition; OPN, osteopontin; PF4+, platelet factor 4.

**Table 8 jpm-13-01259-t008:** The main studies reporting the role of OPN in tuberculosis.

Author/Ref	Study Design	Study Population	Main Findings
Koguchi Y et al. [96]	Observational study	29 male and 19 female patients with active pulmonary tuberculosis.	High plasma OPN concentrations in patients with pulmonary tuberculosis and correlation between plasma OPN levels and clinical parameters in patients with pulmonary tuberculosis.
Wang D et al. [98]	Systematic review and meta-analysis	17 retrospective studies with 933 tuberculosis participants and 786 healthy controls were finally included in this article.	Higher serum/plasma OPN levels were found in tuberculosis patients.
Inomata S-I et al. [100]	Observational cross-sectional study	A total of 47 patients with pulmonary tuberculosis and 7 patients with miliary tuberculosis before anti-tuberculosis therapy, and also measured in 19 patients with tuberculosis before and after anti-tuberculosis therapy.	Circulating IFN-γ, IL-18, and OPN levels were significantly higher in patients with pulmonary tuberculosis than in healthy controls, while there was no significant difference in levels of circulating IL-12 between tuberculosis patients and controls. Moreover, among Th1 response-associated molecules, circulating levels of IL-18 and OPN, but not IFN-γ or IL-12, reflect disease activity in patients with tuberculosis.
Moschos C et al. [101]	Prospective study	A total of 109 consecutive patients with pleural effusions of different etiologies were recruited prospectively during daily clinics.	OPN levels were elevated in exudative pleural effusions, as compared with the levels in blood or transudative pleural effusions. While PF and PF/serum OPN were higher in patients with malignancies.

Abbreviations: IFN-γ, interferon-γ; IL-18, interleukin-18; IL-12, interleukin-12; PF, peripheral blood.

## Data Availability

Not applicable.

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
