# Peer review of "The Role of Osteopontin in Respiratory Health and Disease"

_jpm, 2023, doi:10.3390/jpm13081259_

Round 1

Reviewer 1 Report

First few references are out of order in the manuscript as noted in the attached  annotated manuscript.

Author Response

Response to Reviewer 1 Comments

  • Comment 1: First few references are out of order in the manuscript as noted in the attached annotated manuscript

Response 1: Thank you for the comment. The out of order references in the manuscript have been now corrected. Thank you for this direction.

We appreciate you taking the time to offer us your insights related to the paper.

Reviewer 2 Report

The article attempts to summarize the implications of osteopontin – a well-known but arguably rarely used biomarker over the spectrum of respiratory disorders. This literature review seems generally sound and convincing, and it may be of particular interest to respiratory phisycians. 

Still there are some general suggestions: 

- more details concerning the article selection process would be welcome – a PRISMA flow diagram may be useful 

- the method section would benefit from some introductory phrase aimed to explain the structure of the paper, such as 

..we present data on the normal functions of OPN and its implication in various diseases: asthma, COPD etc 

- the limits of OPN as a biomarker should be also explored in a systematic fashion – in clinical settings OPN is not readily available and used because its relative lack of specificity 

- the conclusions could be improved – possibly indicating future research directions or potential developments 

Furthermore some rephrasing may be necessary – here are some examples 

line 210 should be rephrased to something along the line  

… OPN is higher or up regulated during acute exacerbations of COPD 

line 218 linking OPN at least one of the two major COPD phenotypes – there are more than two major phenotypes, however this could be changed to ….linking OPN to treatable traits of COPD... 

line 261 ...involved in tissue fibrosis and can accelerate the progression of fibrosis through different and not yet fully understood mechanisms; SPP1, the OPN gene, is a highly observed biomarker for IPF and is deposited in fibrotic lungs. At the same time, OPN levels have been proven to be present in IPF patients. 

should be rephrased for clarity – higher OPN levels were present in IPF patients; the gene was over expressed… 

line 474 Moreover, since COVID-19 is a respiratory disease, OPN certainly has a link to it.  

Should be removed or explained as the logic behind it is not clear.

Manuscript should be rechecked for English clarity and readability.

There are some typos: line 227 lug fibrosis and unnecessary capitalizations: lina 210 COPD AND AECOPD

Author Response

Response to Reviewer 2 Comments

Comment 1: The article attempts to summarize the implications of osteopontin – a well-known but arguably rarely used biomarker over the spectrum of respiratory disorders. This literature review seems generally sound and convincing, and it may be of particular interest to respiratory physicians. 

Response 1. We sincerely thank you for the kind words regarding our paper.

Comment 2: Still there are some general suggestions: 

- more details concerning the article selection process would be welcome – a PRISMA flow diagram may be useful 

Response: Thank you for the comment. A flow diagram has been inserted in the paper, as suggested.

- the method section would benefit from some introductory phrase aimed to explain the structure of the paper, such as ..we present data on the normal functions of OPN and its implication in various diseases: asthma, COPD etc 

Response: Thank you for the comment. The Methods of the manuscript has now been enriched with more information regarding the selection process, as recommended.

Comment 3: - the limits of OPN as a biomarker should be also explored in a systematic fashion – in clinical settings OPN is not readily available and used because its relative lack of specificity - the conclusions could be improved – possibly indicating future research directions or potential developments. 

Response 3. Thank you for this direction. You raise a very valid point. We totally agree that there is limited use of OPN in everyday clinical setting and a systemic reference of its limits would improve the quality of the manuscript. Furthermore, future research directions have been mentioned (lines 587-590) while the usefulness of OPN as a biomarker is reported in lines 459-460, 554-558, 315-333, and 599-601.

Comment 4: Furthermore some rephrasing may be necessary

 – here are some examples line 210 should be rephrased to something along the line  

… OPN is higher or up regulated during acute exacerbations of COPD 

 Response:  Thank you for the constructive feedback. We have rephrased the line 210.

Comment 5. line 218 linking OPN at least one of the two major COPD phenotypes – there are more than two major phenotypes, however this could be changed to ….linking OPN to treatable traits of COPD... 

Response: Thank you for this direction. Line 218 has now been changed, as suggested

Comment 7. line 261 ...involved in tissue fibrosis and can accelerate the progression of fibrosis through different and not yet fully understood mechanisms; SPP1, the OPN gene, is a highly observed biomarker for IPF and is deposited in fibrotic lungs. At the same time, OPN levels have been proven to be present in IPF patients. 

should be rephrased for clarity – higher OPN levels were present in IPF patients; the gene was over expressed… 

Response:  Line 261 to has been changed to ‘Higher OPN levels … Therefore, OPN is identified as one of the genes distinguishing IPF from normal lungs. Thank you for the comment.

Comment 8. line 474 Moreover, since COVID-19 is a respiratory disease, OPN certainly has a link to it.  

Should be removed or explained as the logic behind it is not clear.

Response: Thank you for this point, line 474 has now been changed to ‘Since OPN is T – helper cytokine … highly possible’ in order for a better understanding.

Comment 9. There are some typos: line 227 lug fibrosis and unnecessary capitalizations: lina 210 COPD AND AECOPD

Response 9. Thank you for the suggestions, the typos have now been corrected.

We appreciate you taking the time to offer us your insights related to the paper. We hope you find these revisions rise to your expectations.

Reviewer 3 Report

The Authors propose a review of the recent literature involving osteopontin in human diseases of the respiratory system. It is a timely, precise, well written and intelligently reasoned review. The subject has been dealt with rigorously, the conclusions are correct in my opinion. Ultimately I suggest the publication with minimal corrections.

A few issues arise:

1)      The term "bile" is listed 2 times (line 32, 33)

2)      The sentence between lines 44 and 47 is not fluent, please change it

3)      In chapter 8 the role of osteopontin in the context of pulmonary hypertension is examined. A 2021 study demonstrates a possible role of osteopontin in intercepting pulmonary hypertension secondary to immune-mediated connective tissue disease, please consider the citation: Role of Osteopontin as a Potential Biomarker of Pulmonary Arterial Hypertension in Patients with Systemic Sclerosis and Other Connective Tissue Diseases (CTDs).

Author Response

Comment 1. The Authors propose a review of the recent literature involving osteopontin in human diseases of the respiratory system. It is a timely, precise, well written and intelligently reasoned review. The subject has been dealt with rigorously, the conclusions are correct in my opinion. Ultimately I suggest the publication with minimal corrections.

Response 1. Thank you for the kind words. We are delighted to receive positive feedback from you.

Comment 2. The term "bile" is listed 2 times (line 32, 33)

Response 2. Thank you for this point. The change has been made.

Comment 3. The sentence between lines 44 and 47 is not fluent, please change it

Response 3. We thank you for this point, the sentence rephrased.

Comment 4.   In chapter 8 the role of osteopontin in the context of pulmonary hypertension is examined. A 2021 study demonstrates a possible role of osteopontin in intercepting pulmonary hypertension secondary to immune-mediated connective tissue disease, please consider the citation: Role of Osteopontin as a Potential Biomarker of Pulmonary Arterial Hypertension in Patients with Systemic Sclerosis and Other Connective Tissue Diseases (CTDs).

Response 4. Thank you for the comment. The citation has been added,

We appreciate all of your insightful comments. We found them quite useful as we approached our revision. We are grateful for the time and energy you expended on our behalf.